# Artificial Intelligence Applied to Non-Invasive Imaging Modalities in Identification of Nonmelanoma Skin Cancer: A Systematic Review

**DOI:** 10.3390/cancers16030629

**Published:** 2024-02-01

**Authors:** Emilie A. Foltz, Alexander Witkowski, Alyssa L. Becker, Emile Latour, Jeong Youn Lim, Andrew Hamilton, Joanna Ludzik

**Affiliations:** 1Department of Dermatology, Oregon Health & Science University, Portland, OR 97201, USA; 2Elson S. Floyd College of Medicine, Washington State University, Spokane, WA 99202, USA; 3John A. Burns School of Medicine, University of Hawai’i at Manoa, Honolulu, HI 96813, USA; 4Biostatistics Shared Resource, Knight Cancer Institute, Oregon Health & Science University, Portland, OR 97201, USA

**Keywords:** artificial intelligence, dermoscopy, reflectance confocal microscopy, nonmelanoma skin cancer, basal-cell carcinoma, squamous-cell carcinoma, non-invasive imaging, early detection, machine learning

## Abstract

**Simple Summary:**

Artificial intelligence (AI) has shown promise in detecting and diagnosing nonmelanoma skin cancer through image analysis. The incidence of skin cancer continues to rise each year, and it is estimated that one in five Americans will have nonmelanoma skin cancer at some point in their lifetime. Non-invasive diagnostic tools are becoming more widely adopted as the standard of care. When integrated with AI, there is the potential to identify skin cancer earlier and more rapidly compared to traditional methods. This review aims to assess the current status of AI diagnostic algorithms in tandem with noninvasive imaging for the detection of nonmelanoma skin cancer.

**Abstract:**

Background: The objective of this study is to systematically analyze the current state of the literature regarding novel artificial intelligence (AI) machine learning models utilized in non-invasive imaging for the early detection of nonmelanoma skin cancers. Furthermore, we aimed to assess their potential clinical relevance by evaluating the accuracy, sensitivity, and specificity of each algorithm and assessing for the risk of bias. Methods: Two reviewers screened the MEDLINE, Cochrane, PubMed, and Embase databases for peer-reviewed studies that focused on AI-based skin cancer classification involving nonmelanoma skin cancers and were published between 2018 and 2023. The search terms included skin neoplasms, nonmelanoma, basal-cell carcinoma, squamous-cell carcinoma, diagnostic techniques and procedures, artificial intelligence, algorithms, computer systems, dermoscopy, reflectance confocal microscopy, and optical coherence tomography. Based on the search results, only studies that directly answered the review objectives were included and the efficacy measures for each were recorded. A QUADAS-2 risk assessment for bias in included studies was then conducted. Results: A total of 44 studies were included in our review; 40 utilizing dermoscopy, 3 using reflectance confocal microscopy (RCM), and 1 for hyperspectral epidermal imaging (HEI). The average accuracy of AI algorithms applied to all imaging modalities combined was 86.80%, with the same average for dermoscopy. Only one of the three studies applying AI to RCM measured accuracy, with a result of 87%. Accuracy was not measured in regard to AI based HEI interpretation. Conclusion: AI algorithms exhibited an overall favorable performance in the diagnosis of nonmelanoma skin cancer via noninvasive imaging techniques. Ultimately, further research is needed to isolate pooled diagnostic accuracy for nonmelanoma skin cancers as many testing datasets also include melanoma and other pigmented lesions.

## 1. Background

Nonmelanoma skin cancer, primarily basal-cell and squamous-cell carcinoma types, is the most common cutaneous malignancy, accounting for 98% of skin cancers diagnosed [1]. Early detection of skin cancer can reduce morbidity by up to 90% [2]. Traditional skin cancer diagnostic methods can be costly, take time, have potential for resource limitation, and require a well-trained dermatology provider. Non-invasive tools used for diagnosis are increasingly prevalent as a standard of care, particularly for patients with an extensive history of skin cancer. These techniques combined with the application of AI can detect skin cancer early. Thus, AI tools are being increasingly used, including shallow and deep machine learning–based methodologies that are trained to detect and classify skin cancer using computer algorithms and deep neural networks [3]. However, to date, no AI algorithms have been Food and Drug Administration (FDA) cleared (Class II) in the field of dermatology.

As AI becomes increasingly integrated into all computerized functions of medicine and daily activities, it is essential to recognize its potential to assist in computer-directed diagnostics. Utilizing AI, systems can analyze images of skin lesions, pinpointing features indicative of nonmelanoma skin cancer. These systems employ deep learning and convolutional neural networks (CNNs) to train algorithms on extensive datasets of labeled images [4]. An advantage of AI-based nonmelanoma skin cancer imaging lies in its potential for more precise and efficient diagnoses. Dermatology providers can swiftly assess images using AI-based systems, identifying suspicious lesions for further evaluation. Additionally, AI-based nonmelanoma skin cancer imaging holds promise in reducing the necessity for unnecessary, invasive, and costly biopsies. By accurately identifying potentially cancerous lesions, AI-based systems empower dermatologists to target biopsies toward the most concerning areas within a skin lesion.

Numerous studies have proposed innovative designs for skin cancer identification through image analysis [5]. Over time, there has been a growth in computational capabilities through novel and existing approaches, along with expanded datasets for interpretation, leading to robust mathematical models in the current state of AI in the field. Various entities are developing their own AI algorithms for diverse diagnostic modalities and assessing their accuracy [5].

AI is a comprehensive term encompassing computer-aided automated decision-making and is increasingly applied across various aspects of medicine. Machine learning (ML), a subset of AI, involves the use of technologies for data prediction. Subcategories include shallow and deep learning. Both shallow and deep machine learning methods have been trained to identify and classify skin cancer, with algorithms designed to predict malignancies based on patterns found in large datasets of skin lesion images gaining prominence.

This review investigates the utilization of various machine learning mechanisms for non-invasive image analysis. Before delving into our analysis, it is crucial to establish clear definitions for the common terms that will be referenced throughout the discussion.

### 1.1. Common Machine Learning Methods

As noted above, deep learning is a category of machine learning. This type of algorithm uses machines to interpret and manipulate data from images, speech, or language. Deep learning can be further subcategorized into different types of neural networks. A CNN, or convolutional neural network, is a specialized form of deep neural network (DNN) designed for processing image data. Comprising multiple layers, including convolutional layers, pooling layers, and fully connected layers, CNNs are tailored to efficiently learn features within images [6]. On the other hand, a DNN is a broader category with multiple layers, typically exceeding three, and finds applications in various domains such as image classification, speech recognition, and natural language processing. The key distinction between CNNs and DNNs lies in their approach to processing image data. CNNs are optimized for feature learning in images, employing convolution techniques to extract patterns by sliding a small filter over the image and computing dot products with pixels. DNNs, in contrast, often use fully connected layers for image processing, linking every neuron in one layer to every neuron in the next, resulting in a larger number of parameters that need optimization during training [6].

Lastly, a deep convolutional neural network (DCNN) is a subtype of CNN with additional layers, enabling it to learn more intricate features and patterns in data. This enhancement contributes to superior performance in tasks like image classification and object detection. The primary difference between CNNs and DCNNs lies in the number of layers, with DCNNs potentially having dozens or even hundreds. While DCNNs offer heightened accuracy, they demand more computational resources and training data, making them more susceptible to overfitting.

### 1.2. AI Applications in Non-Invasive Imaging Modalities

AI has shown potential in improving the accuracy of nonmelanoma skin cancer diagnosis using dermoscopy and reflectance confocal microscopy (RCM) [3]. Dermoscopy is a non-invasive imaging technique that uses a handheld device to magnify and illuminate skin lesions [1]. AI-based systems can analyze dermoscopy images and identify patterns and features that are indicative of nonmelanoma skin cancer. For example, an AI algorithm can be trained to detect the presence of specific structures, such as white lines, dots, and vascular structures—that are associated with nonmelanoma skin cancer. One advantage of using AI in dermoscopy is the potential for more accurate and efficient diagnoses [5]. Dermoscopy outcomes can be highly user-dependent, leading to variability and poor reproducibility. Applying pattern recognition in deep learning to dermoscopic images can address this concern.

RCM is a non-invasive imaging technique that allows dermatologists to examine skin lesions at a cellular level. It allows in vivo visualization of skin lesions at a near-histological resolution [5]. It employs a diode laser, and captures horizontal images that are as superficial as the stratum corneum and as deep as the upper dermis. AI-based systems can analyze RCM images to identify patterns and features that are indicative of nonmelanoma skin cancer. For example, an AI algorithm can be trained to detect the presence of abnormal cells, blood vessels, and other features that are characteristic of nonmelanoma skin cancer. On RCM images, numerous studies have applied AI to automatically localize and classify layers of the epidermis [5]. Additional studies have used ML in the detection of the dermal-epidermal junction (DEJ), allowing for immediate visualization of potential malignant features in the DEJ. Applying AI to RCM in skin cancer detection has potential for more reproducible and consistent interpretations of skin architecture. Challenges include diminished image quality due to large RCM files, increased cost and resources, and decreased variability phenotypically. Dermatologists can use AI-based systems to quickly analyze images and identify suspicious lesions that require further evaluation. Additionally, AI-based systems can help reduce inter-observer variability and increase diagnostic accuracy by providing an objective assessment of images.

In this literature review, we sought to collect the latest machine learning algorithms that are being applied to non-invasive diagnostic techniques in nonmelanoma skin cancers. Many algorithms predict the malignancy of pigmented lesions in skin cancer; however, the diagnosis of non-pigmented lesions is generally considered more challenging. To our knowledge, this literature review is the first of its kind to isolate and describe the current state of AI in non-invasive imaging modalities’ ability to accurately classify nonmelanoma skin cancers.

## 2. Materials and Methods

### 2.1. Search Strategy

Articles published from January 2018–December 2023 were identified from comprehensive searches of MEDLINE, Cochrane, and Embase. Search terms included “skin neoplasms”, “diagnostic techniques and procedures”, “artificial intelligence”, “algorithms”, “computer systems”, “lesion or growth or cancer or neoplasm or tumor or malignant or metastatic”, “carcinoma”, “machine or deep learning”, “neural network”, “diagnosis or detection”, “nonmelanoma”, “basal-cell carcinoma”, “squamous-cell carcinoma”, “dermoscopy”, “reflectance confocal microscopy”, and “optical coherence tomography”. Records were screened from MEDLINE, Cochrane, Embase, and Pubmed databases, yielding a total of 967 articles. Prior to initial screening, duplicate articles and articles published prior to 2018 were excluded.

### 2.2. Study Selection

Preferred Reporting Items for Systematic Reviews and Meta-analyses (PRISMA) guidelines were followed throughout this study. The protocol has not been registered. Search results were evaluated by two independent reviewers, and in the case of a discrepancy in study selection or inclusion criteria, a third reviewer was involved for resolution. Only original, peer-reviewed research manuscripts in the English language were selected for review. We subsequently screened the articles through a review of the title and abstract, with consideration of the research question and appropriate inclusion and exclusion criteria. A total of 317 records were reviewed as full-text articles and considered for inclusion in this review based on our defined inclusion and exclusion criteria. Inclusion criteria included (i) discussion of a novel machine learning algorithm proposal or design, (ii) numerical outcomes reporting the algorithm’s accuracy, (iii) an algorithm that completes all steps to diagnosis (not stopping at segmentation, but proceeding to classification), (iv) the study population being human subjects, (v) publication in English, and (vi) the full text being available (Figure 1). Exclusion criteria included articles that (i) failed to address our research question, (ii) utilized invasive techniques for diagnosis, and (iii) screened only based on clinical images (without the use of additional advanced imaging tools).

### 2.3. Study Analysis

This review systematically evaluated the effectiveness of AI-based methodologies in conjunction with reflectance confocal microscopy, optical coherence tomography, and dermoscopy for detecting nonmelanoma skin cancers. Thus, we elucidated performance metrics including accuracy, sensitivity, specificity, area under the curve (AUC), positive predictive value (PPV), and negative predictive value (NPV). The term “accuracy” refers to the percentage of lesions correctly classified, while “sensitivity” and “specificity” quantify the proportions of true positive and true negative cases, respectively. The AUC comprehensively summarizes the overall performance of the classification model, while the PPV and NPV describe the proportions of lesions accurately reflecting the presence or absence of nonmelanoma skin cancer. By scrutinizing and comparing these performance metrics, we summarized the effectiveness of AI-applied nonmelanoma skin cancer using noninvasive imaging modalities.

## 3. Results

A total of forty-four articles were selected for review by means of fulfilling our inclusion criteria. Twenty-six of the forty-four articles were published in 2022, five in 2021, and ten in 2020. Prior to 2020, only three articles that were published met our inclusion criteria (Figure 2).

The majority of the articles described machine learning algorithms used to interpret dermoscopic images (*n* = 38). Of note, each study utilized variable metrics to quantify the performance of the algorithm. Such metrics included accuracy, precision, sensitivity, specificity, positive predictive value (PPV), negative predictive value (NPV), AUC (area under the curve), and F1 score (Table 1). In each of the reported metrics, a high percentage is correlated with superior algorithm performance.

The most frequently recorded performance metric across studies included in Table 1 was accuracy. Hosny et al.’s convolutional neural network boasted an accuracy of 98.70% [7], which was the highest yielding accuracy of dermoscopy algorithms. Dermoscopy-applied AI detection of NMSC yielded an average accuracy of 86.80% with a standard deviation of 12.05%, a median accuracy of 90.54%, and a minimum accuracy of 37.6%.

Three articles used novel algorithms in association with RCM images, and one used AI applied to hyperspectral epidermal imaging (HEI) (Table 2). Each of the studies applying AI to RCM images reported algorithmic efficacy via different metrics. Wodzinski et al. yielded an accuracy of 87%, Chen et al. reported sensitivity and specificity yielding 100% and 92.4%, respectively, and Campanella et al. recorded an AUC of 86.1% [8,9,10]. HEI (La Salvia et al.) yielded outcome measures of 87% sensitivity, 88% specificity, and an AUC of 90%, though no reported accuracy [11].

The utilization of diverse image databases for the analysis of AI algorithms showcased additional variability among study designs. Table 3 provides a detailed overview of public dermoscopy image databases that were utilized by studies included in the review.

Lastly, Figure 3 displays the variety of machine learning methods utilized across the studies included in our systematic review. The majority of the studies used CNN as the method for the generation of their machine learning algorithms, with deep learning as the second most common method. DCNNs and DNNs were utilized by a small number of studies, and each of the other papers applied novel, independently generated methods in their algorithms.

Given the diverse variability of the study design for each included study, a pooled analysis was not able to be calculated. Rather, a QUADAS-2 risk of bias assessment was performed (Table 4). Per QUADAS-2 guidelines, both risk of bias and applicability concerns were evaluated in subcategories including patient selection, index test, reference standard, and flow and timing [12]. None of the studies demonstrated a high risk of bias in any category. However, 11 studies demonstrated high risk of applicability concerns in regard to the index test. No other studies demonstrated high risk of applicability concerns in other categories.

**Table 1 cancers-16-00629-t001:** Summary of included studies utilizing dermoscopy images.

Authors	Image Dataset	Accuracy	Precision	Sensitivity	Specificity	PPV	NPV	AUC	F1 Score
Hosny et al. (2020) [7]	Internal dataset	98.7%	95.1%	95.6%	99.3%				
Xin et al. (2022) [13]	HAM1000	94.3%	94.1%						
Xin et al. (2022) [13]	Internal dataset	94.1%	94.2%						
Tang et al. (2022) [14]	Seven Point Checklist	74.9%							
Skreekala et al. (2022) [15]	HAM1000	97%							
Sangers et al. (2022) [16]	HAM1000			86.9%	70.4%				
Samsudin et al. (2022) [17]	HAM1000	87.7%							
S M et al. (2022) [18]	ISIC 2019 & 2020							96.8% *	
Reis et al. (2022) [19]	ISIC 2018	94.6%							
Reis et al. (2022) [19]	ISIC 2019	91.9%							
Reis et al. (2022) [19]	ISIC 2020	90.5%							
Razzak et al. (2022) [20]	ISIC 2018	98.1%							
Qian et al. (2022) [21]	HAM1000	91.6%		73.5%	96.4%			97.1%	
Popescu et al. (2022) [22]	ISIC 2018	86.7%							
Nguyen et al. (2022) [23]	HAM1000	90%	81%					99%	81%
Naeem et al. (2022) [24]	ISIC 2019	96.9%							
Li et al. (2022) [25]	HAM1000	95.8%	96.1%						95.7%
Lee et al. (2022) [26]	ISIC 2018	84.4%		92.8%		78.5%	91.2%		
Laverde-Saad et al. (2022) [27]	HAM1000	77.1%		80%	86%	86%	80%		
La Salvia et al. (2022) [28]	HAM1000			>80%	>80%			>80%	
Hosny et al. (2022) [29]	Several datasets	94.1% *	91.4% *	91.2% *	94.7% *				
Dascalu et al. (2022) [30]	Internal dataset	88%		95.3%				91.1%	
Combalia et al. (2019) [31]	HAM1000	58.8%							
Benyahia et al. (2022) [32]	ISIC 2019	92.3%							
Bechelli et al. (2022) [33]	HAM1000	88%							
Bechelli et al. (2022) [33]	HAM1000	72%							
Afza et al. (2022) [34]	Ph2	95.4%							
Afza et al. (2022) [34]	ISBI2016	91.1%							
Afza et al. (2022) [34]	HAM1000	85.5%							
Afza et al. (2022) [35]	HAM1000	93.4%							
Afza et al. (2022) [35]	ISIC2018.	94.4%							
Winkler et al. (2021) [36]	HAM1000	70%		70.6%	69.2%				
Pacheco et al. (2021) [37]	HAM1000	77.1%							
Minagawa et al. (2021) [38]	HAM1000	85.3%							
Iqbal et al. (2021) [39]	HAM1000							99.1%	
Huang et al. (2021) [40]	HAM1000	84.8%							
Zhang et al. (2020) [41]	DermIS & Dermquest	91%		95%	92%	84%	95%		
Wang et al. (2020) [42]	Several datasets			80%	100%				
Qin et al. (2020) [43]	HAM1000	95.2%		83.2%	74.3%				
Mahbod et al. (2020) [44]	ISIC2019	86.2%							
Li et al. (2020) [45]	HAM1000		78%	95%	91%	87%			
Gessert et al. (2020) [46]	HAM1000			70%					
Gessert et al. (2020) [47]	Internal dataset			53% *	97.5% *			94% *	
Al-masni et al. (2020) [48]	HAM1000	89.3%							
Ameri et al. (2020) [49]	HAM1000	84%		81%					
Tschandl et al. (2019) [50]	Internal dataset	37.6%		80.5%					
Dascalu et al. (2019) [51]	HAM1000			91.7%	41.8%	59.9%		81.4%	

Items with an asterisk (*) represent averaged values. Shaded boxes indicate that a specific measure was not collected in the study.

**Table 2 cancers-16-00629-t002:** Summary of included studies utilizing imaging modalities other than dermoscopy.

Authors	Imaging Modality	Accuracy	Sensitivity	Specificity	AUC
Wodzinski et al. (2019) [8]	RCM	87%			
Chen et al. (2022) [9]	RCM		100% (when combined with RS)	92.4% (when combined with RS)	
Campanella et al. (2022) [10]	RCM				86.1%
La Salvia et al. (2022) [11]	HEI		87%	88%	90%

Shaded boxes indicate that a specific measure was not collected in the study.

**Table 3 cancers-16-00629-t003:** Description of databases tested.

Database	Image Type	Total Images	Description of Dataset
HAM1000	Dermoscopy	10,015	Melanoma (MM)—1113 images Vascular—142 images Benign nevus (MN)—6705 images Dermatofibroma (DF)—115 images Seborrheic keratosis (SK)—1099 Basal-cell carcinoma (BCC)—514 images Actinic keratosis (AK)—327 images
Xin et al. [13] Internal	Dermoscopy	1016	BCC—630 images Squamous-cell carcinoma (SCC)—192 images MM—194 images
SPC	Dermoscopy	>2000	MM, BCC, SK, DF, solar lentigo (SL), vascular, SK Note: Distribution of number of images per lesion type varies in the literature.
ISIC 2016	Dermoscopy	1279	Distribution of number of images per lesion type not readily available
ISIC 2017	Dermoscopy	2000	MM—374 images SK—254 images Other/unknown—1372 images
ISIC 2018	Dermoscopy	10,015	MM—1113 images MN—6705 images BCC—514 images AK—327 images SK—1099 images DF—115 images Vascular—142 images
ISIC 2019	Dermoscopy	25,331	MM—4522 images MN—12,875 images BCC—3323 images AK—867 images DF—239 images SK—2624 images SCC—628 images Vascular—253 images
ISIC 2020	Dermoscopy	33,126	MM—584 images AMN—1 image Café-au-lait macule—1 image SL—44 Lichenoid keratosis—37 images Other/unknown—27124 images
PH2	Dermoscopy	200	Not available

**Table 4 cancers-16-00629-t004:** Summary of QUADAS-2 analysis.

Categories	Risk of Bias				Applicability Concerns		
	Patient Selection	Index Test	Reference Standard	Flow and Timing	Patient Selection	Index Test	Reference Standard
Low Risk	27/44	31/44	42/44	44/44	20/44	37/44	44/44
High Risk	0/44	0/44	0/44	0/44	10/44	0/44	0/44
Unclear/Moderate	17/44	13/44	2/44	0/44	14/44	7/44	0/44

## 4. Discussion

AI facilitates more accurate triage and diagnosis of skin cancer through digital image analysis, empowering dermatologists [52]. Various techniques, including machine learning, deep learning, and CNNs, are employed in AI-based skin cancer detection. These methods utilize labeled image datasets to train algorithms, enabling them to recognize patterns and features indicative of skin cancer in lesions [2].

AI exhibits significant promise in the detection of skin cancer, yet ongoing efforts to optimize its potential are evident in the trajectory of publication years. The decline in publications in 2021 may be attributed to pandemic-related limitations on resources and the ability to generate novel machine learning algorithms.

Compared to traditional methods, AI-based skin cancer detection offers several advantages. Firstly, AI algorithms swiftly analyze large image datasets, providing dermatologists with more accurate and timely diagnoses. Secondly, these systems reduce the necessity for unnecessary and invasive biopsies, cutting down on costs. Thirdly, AI-based systems can be deployed in remote or underserved areas where access to dermatologists is limited [1].

The reported average diagnostic accuracy of 86.80% when AI is applied to dermoscopic images and a diagnostic accuracy of 87% for RCM-based AI are promising indicators of automated image interpretation potential. However, the wide standard deviation and variability between the minimum accuracy of 37.6% and a maximum of 98.7% in AI applied to dermoscopy underscore the need for further standardization and broader accuracy improvement efforts.

There is a lack of literature on the application of AI to OCT in human lesions. While Ho et al. utilized deep learning for SCC detection in mice, achieving 80% accuracy, there are currently no AI algorithms in the literature for detecting NMSC via OCT in humans [53].

### 4.1. Limitations

Our QUADAS-2 assessment of bias demonstrates that “patient selection” was unsatisfactory in many of the studies included in this review. This is because the images tested and trained on these AI models frequently utilized public databases of dermoscopy images. Many of these datasets accessible to the public have insufficient sample sizes, thus impacting an AI algorithm’s ability to train and reprogram itself [12].

Moreover, imbalanced datasets pose a common challenge for AI models, especially in supervised machine learning where the algorithm is trained on labeled data. Imbalanced datasets arise when there is an unequal distribution of examples among different classes, leading to a skewed representation of certain classes compared to others. For instance, in the context of skin cancer, variations in the incidence of each skin cancer type and a higher percentage of the population with no skin cancer (referred to as “healthy” individuals) contribute to imbalanced datasets. If the training data for the AI model predominantly consists of healthy individuals, it may struggle to accurately predict rarer diseases due to the lack of relevant examples [54].

The primary drawback of imbalanced datasets for AI models is their potential to produce biased and inaccurate results. The model might exhibit a bias toward the majority class, leading to subpar performance on the minority class. In extreme cases, the model might disregard the minority class entirely, resulting in complete misclassification.

In the classification of skin cancer images, this imbalance can be particularly problematic for individuals with darker skin tones, as there is insufficient diversity in skin tone inputs. Existing AI models have mainly been trained on European or East Asian populations, and the limited representation of darker skin tones may compromise overall diagnostic accuracy. This can introduce bias toward Fitzpatrick skin types 4–6, making the model less adept at recognizing or interpreting images of individuals with darker skin tones compared to those with lighter skin tones [55]. Additionally, AI models may rely on color contrast as a pivotal factor in image interpretation, which could lead to misinterpretation due to lower contrast between darker skin tones and other colors compared to lighter skin tones. These limitations carry significant implications for the accuracy and fairness of AI applications across various fields. Therefore, it is essential to ensure that AI models undergo training on diverse datasets and are systematically tested for biases to ensure accurate results and equitable access to emerging health technologies [56].

Furthermore, the efficacy of AI is heavily influenced by image quality, and various factors contribute to variability in this aspect. Differences in image acquisition and quality present a barrier to the implementation of AI in the clinical setting that must be overcome. Achieving consistent, high-quality images necessitates addressing issues such as artifact removal (e.g., hairs, surgical ink markings) and ensuring attention to zoom level, focus, and lighting.

### 4.2. Future Directions

Future directions may consider automated identification of pigmented lesions, detection of different architectural patterns to distinguish malignant versus benign lesions, categorization of lesions as melanoma versus nonmelanoma skin cancer, and identification of individual skin cells or nuclei using machine learning technologies. It is important to note that the application of AI in dermatology is not a threat to a dermatologist’s livelihood—it can be an asset. AI does not devalue the utility of dermatologists, but rather enables a better allocation of their time. Redirecting this finite time can allow for more time spent with patients, increase accessibility to dermatologists, and may increase the accuracy and reproducibility of non-invasive imaging techniques.

## 5. Conclusions

Overall, AI has the potential to revolutionize the field of skin cancer detection by improving diagnostic accuracy and reproducibility, leading to earlier detection and better outcomes for patients. A high risk of bias and applicability concerns was observed in several of the included studies analyzed via QUADAS-2 assessment. Furthermore, a moderate risk of bias and applicability concerns was observed among many studies. This indicates a need for further standardized evaluation metrics to reduce these biases in studies evaluating diagnostic accuracy. It is also important to note that AI-based skin cancer detection is still in its early stages, and more research is needed to fully evaluate its accuracy and effectiveness, as well as to streamline measures of efficacy. Lastly, AI-based systems should be used as an adjunct stand-alone tool to support dermatologists in their diagnosis rather than as a replacement for human expertise. Ultimately it is the responsibility of the dermatology provider to make an independent decision on how to properly manage their own patients while considering the ancillary information provided by the use of technology such as AI.

## Figures and Tables

**Figure 1 cancers-16-00629-f001:**
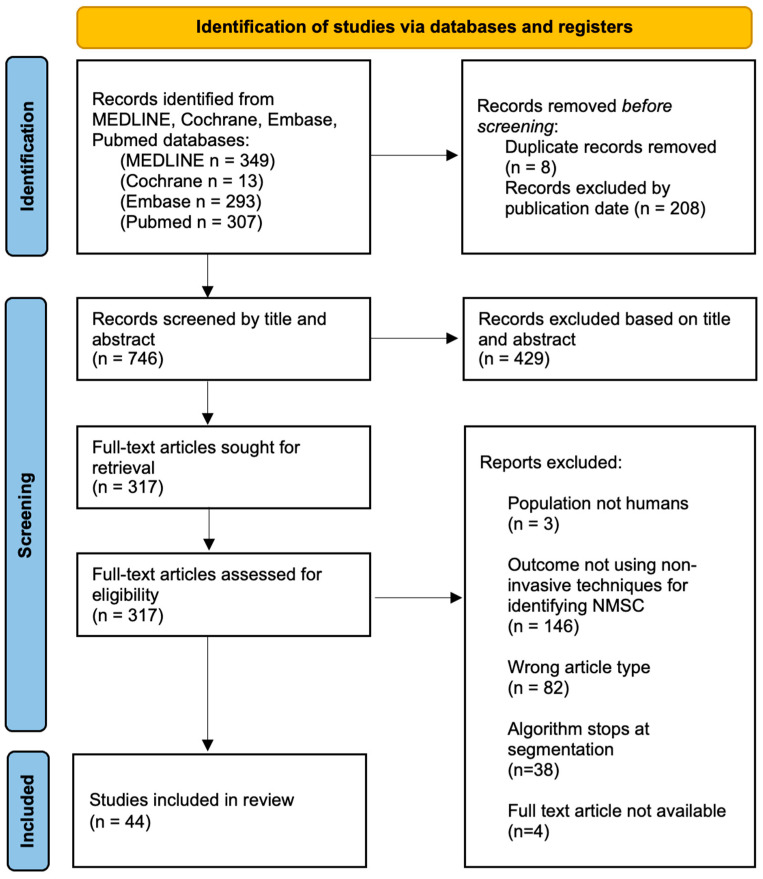
PRISMA 2020 flow diagram for new systematic reviews including searches of databases.

**Figure 2 cancers-16-00629-f002:**
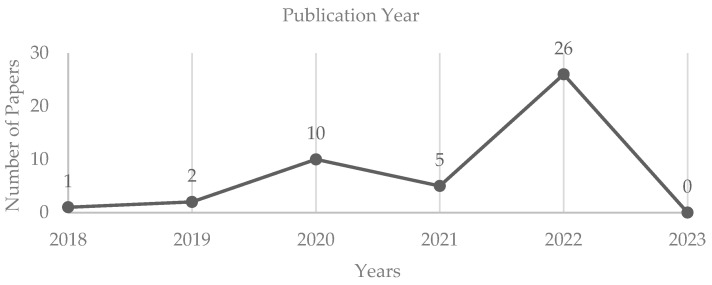
The number of papers per year published included in the literature review.

**Figure 3 cancers-16-00629-f003:**
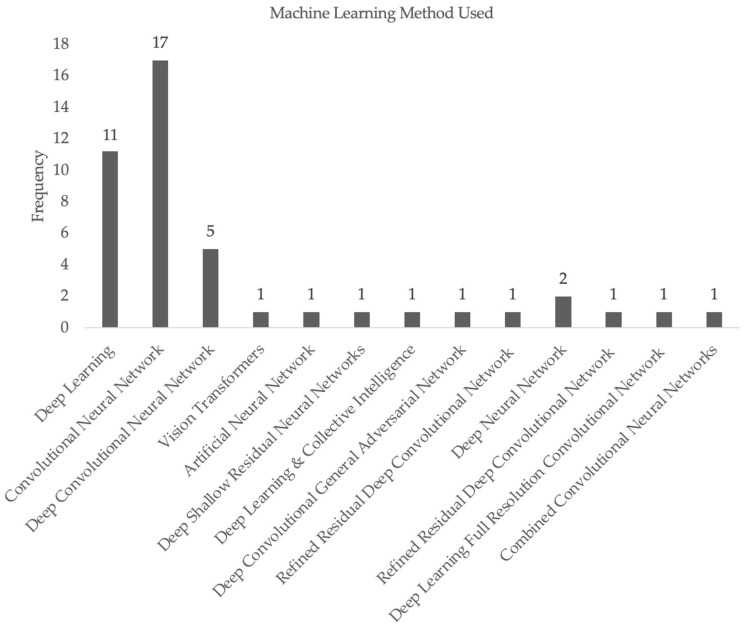
Frequency of machine learning techniques used for papers included in our systematic review.

## Data Availability

Data presented in this study available on request.

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
