# Peer review of "Artificial Intelligence Applied to Non-Invasive Imaging Modalities in Identification of Nonmelanoma Skin Cancer: A Systematic Review"

_cancers, 2024, doi:10.3390/cancers16030629_

Round 1
Reviewer 1 Report
Comments and Suggestions for Authors
In this study, the authors aim to systematically analyze the current state of the literature regarding novel AI models utilized in non-invasive imaging for the early detection of non-melanoma skin cancers. As a review, the description of how to select articles is too long, and there is no more systematic and specific summary and comparison of each method, dataset, etc., which does not serve the purpose of a review article. Overall, there are too few references for a review article.
Besides, there are still some details to be worked out.
1) Figure 1 is not clear.
2) Confusing use of acronyms and full names in the article, e.g., AI in some places and artificial intelligence in others.
3) The background section has almost no references, which does not match the role of this part. 2.1 describes the AI-based approach in non-melanoma skin cancer diagnosis, but 2.2 is an introduction to ML; the logical order of section 2 is confusing.
Comments on the Quality of English LanguageExtensive editing of English language is required.
Author Response
In this study, the authors aim to systematically analyze the current state of the literature regarding novel AI models utilized in non-invasive imaging for the early detection of non-melanoma skin cancers. As a review, the description of how to select articles is too long, and there is no more systematic and specific summary and comparison of each method, dataset, etc., which does not serve the purpose of a review article. Overall, there are too few references for a review article.
Besides, there are still some details to be worked out.
Dear reviewer, thank you for your time in providing us with this feedback. We have made several revisions to our manuscript that you will find attached.
Given the novelty of this area of research, there are limited publications that meet the criteria for inclusion in our systematic review. We feel that the number of citations is appropriate for this reason. Another of our systematic reviews was published with 40 citations, and we consider it the sister piece to this study. It can be found at: doi:10.3390/cancers15194694
1) Figure 1 is not clear.
Thank you for pointing this out. We reviewed and updated the PRISMA diagram (Figure 1) so that it is of higher quality.
2) Confusing use of acronyms and full names in the article, e.g., AI in some places and artificial intelligence in others.
Thank you. All inconsistencies with acronyms have been addressed.
3) The background section has almost no references, which does not match the role of this part. 2.1 describes the AI-based approach in non-melanoma skin cancer diagnosis, but 2.2 is an introduction to ML; the logical order of section 2 is confusing.
This feedback has been taken and applied to our paper. We have merged the background and introduction sections, per reviewer recommendations, and have provided a more logical order of the subsections.
Reviewer 2 Report
Comments and Suggestions for Authors
Artificial Intelligence Applied to Non-Invasive Imaging Modalities in Identification of Nonmelanoma Skin Cancer: A Systematic Review
by: Emilie A. Foltz, Alexander Witkowski, Alyssa L. Becker BS, Emile Latour, Jeong Youn Lim, Andrew Hamilton, Joanna Ludzik,
I have following Observations:
It seems that the manuscript is first draft
* Starting of paper is unusual manner
1 Paper started with section "Simple Summary".
2 Abstract contains major sub sections namely, Background, Methods, Results, Conclusions. Although Background and conclusion are discussed separately.
3. In keywords a single word used "confocal", doyo mean "confocal microscopy" which is already presented after it? and "early detection"! of what?
4. Table 1. In last column: single number #, number with % sign, "precision/accuracy- #%" different formats are used.
5. This study is based on very few papers of last 6 years, the literature review consist on 2018-2023. They have referred 46 studies (42 utilizing dermoscopy, 3 using reflectance confocal microscopy (RCM), and 1 for hyperspectral epidermal imaging). This study should be extended for better results.
6. The detail/discussion about QUADAS-2 is missing only one reference [9] is given.
Author Response
It seems that the manuscript is first draft
Thank you for taking the time to provide feedback regarding our manuscript. We have reviewed all reviewers’ suggestions and applied the recommended revisions accordingly.
* Starting of paper is unusual manner
1 Paper started with section "Simple Summary".
Thank you for this feedback. We referenced the “submission checklist” recommended for the Cancers journal and utilized the template provided. Per their instructions for authors, we followed this format and chose to retain it. It can be found at this link: https://www.mdpi.com/journal/cancers/instructions
2 Abstract contains major sub sections namely, Background, Methods, Results, Conclusions. Although Background and conclusion are discussed separately.
Thank you for this feedback. We referenced the “submission checklist” recommended for the Cancers journal and utilized the template provided for the abstract. Per their instructions for authors, we followed this format and chose to retain it. It can be found at this link: https://www.mdpi.com/journal/cancers/instructions
However, we have merged the background and introduction sections, per reviewer recommendations, and have provided a more logical order of the sections.
- In keywords a single word used "confocal", doyo mean "confocal microscopy" which is already presented after it? and "early detection"! of what?
Dear reviewer, thank you for your feedback. We removed the redundancy of keywords with the extra term “confocal”.
We use the term “early detection” in respect to early detection of cancer, specifically skin cancer, in this manuscript. We have chosen to retain the keywords “early detection” as we consider this an important subsection of cancer research.
- Table 1. In last column: single number #, number with % sign, "precision/accuracy- #%" different formats are used.
This is excellent feedback; we appreciate your attention to detail in pointing it out. We have edited Table 1 for consistency to address this concern.
- This study is based on very few papers of last 6 years, the literature review consist on 2018-2023. They have referred 46 studies (42 utilizing dermoscopy, 3 using reflectance confocal microscopy (RCM), and 1 for hyperspectral epidermal imaging). This study should be extended for better results.
Thank you for noting this. We expanded our search to include all papers up to this date and included those that met our inclusion criteria. Given the novelty of this area of research, we did not expect many publications to exist. Our primary goal is to summarize the progress (however much or little) this area of medicine has made within the last several years.
- The detail/discussion about QUADAS-2 is missing only one reference [9] is given.
Thank you for this comment. Reference 9 (now changed to reference 1 given additional revisions) is provided in the results section of this manuscript. We further discussed QUADAS-2 in our discussion section, primarily under limitations (section 4.1).
Reviewer 3 Report
Comments and Suggestions for Authors
The paper "Artificial Intelligence Applied to Non-Invasive Imaging Modalities in Identification of Nonmelanoma Skin Cancer: A Systematic Review" tries to give a structured overview of AI applied for skin cancer detection. It is well written and structured, but leaves space for improvement in the latter parts.
In Abstract:
- line 32, QUADAS-2 add reference
1. Introduction:
- line 58: introduce abbr. "FDA cleared"
3.1 Search Strategy
- line 161: add space in of967
3.2 Study Selection
- line 166-167: drop designation of reviewer names. Not relevant for the outcome and distracting
Up to here, in my opinion, this work is so well planned, well structured, and well written!
4. Results
- This way, the table (1) is not an asset for this article. It is difficult to get a meaningful overview here, to compare or to see what is remarkable.
- Entries contain redundant information. i.e.: 1st entry: dataset and outcome measures is both repeated in last column. You would increase readability using two rows instead of one (one for each dataset), drop outcome measures column as you name it in last column anyway.
(This might also reduce the irritations generated i.e. within the entry of S M et. al (2022): two datasets, two outcome measures, *one* result value?; Reis et. al (2022) Data "Misc", however, datasets are named in last column, where previously outcome measures were repeated.)
- Is the imaging modality an fixed attribute of the dataset? In that case this could be placed with the dataset. Or drop anyway and split into several tables, this would also increase comparability.
- Qian et al. (2022) two outcome measures named, three results given. No sorry, this is only poor readability of the table.
- Hosney et al. (2022) giving the averages only, also gives the feeling of an loss of information.
- Chen et al. 2022: what is RS? And what if not combined?
- Bechelli et al : what is DL (Deep Learning?) what ML (general Machine Learning Models?) this gets not clear. Either use abbreviations in the table or don't, but keeps consistent with this decisions and make sure abbr. are clear.
- avoid linebreaks within rows.
What do I learn from the table, where is the benefit? Comparability between methods gets completely lost.
So. This way, dropping up to 3 columns would perhaps allow for a restructuring of the table. Give a column for each measurement outcome result (column names vertical) (perhaps even use landscape orientation) and order by i.e. data set. This way the table would at least allow for comparison of different applications performances on the same dataset.
- I would drop figure 2 as it does not carry important information that could not be statet in 1-2 sentences.
- Line 228: it does not get clear which one are those 11 studies
Table 3: Put this table before table 1, and then drop the image type column in table 1.
Also please have a closer look at the table layout: some rows are split over pages which results in poor readability, while other entries have so much space above and beneath.
5. Discussion again is written nicely.
However, this article could be improved so much by having a closer look. Improve the table for better comparability. I.e. Table 1: skreekala / sangers / samsudin: direct comparison is possible at least between two of them since they're close, so one could state that samsudin might be better than skreekala. BUT what is the difference between them? And is the only difference between them and sangers et al. the outcome measure used?
And maybe their work is based on the work of combalia et al. - what made the improvement in those three years between those works?
As given here, the overview is shallow.
Author Response
The paper "Artificial Intelligence Applied to Non-Invasive Imaging Modalities in Identification of Nonmelanoma Skin Cancer: A Systematic Review" tries to give a structured overview of AI applied for skin cancer detection. It is well written and structured, but leaves space for improvement in the latter parts.
Thank you for the time taken to review our manuscript. We have reviewed your recommendations and several revisions have been made accordingly.
In Abstract:
- line 32, QUADAS-2 add reference
Reference has been added.
- Introduction:
- line 58: introduce abbr. "FDA cleared"
Thank you. This has been addressed.
3.1 Search Strategy
- line 161: add space in of967
Thank you for this catch. We have updated accordingly.
3.2 Study Selection
- line 166-167: drop designation of reviewer names. Not relevant for the outcome and distracting
Reviewer names have been removed.
Up to here, in my opinion, this work is so well planned, well structured, and well written!
Thank you for this feedback. We have addressed your concerns in the latter parts of this manuscript and included several revisions to improve the quality of these sections.
- Results
- This way, the table (1) is not an asset for this article. It is difficult to get a meaningful overview here, to compare or to see what is remarkable.
- Entries contain redundant information. i.e.: 1st entry: dataset and outcome measures is both repeated in last column. You would increase readability using two rows instead of one (one for each dataset), drop outcome measures column as you name it in last column anyway.
(This might also reduce the irritations generated i.e. within the entry of S M et. al (2022): two datasets, two outcome measures, *one* result value?; Reis et. al (2022) Data "Misc", however, datasets are named in last column, where previously outcome measures were repeated.)
- Is the imaging modality an fixed attribute of the dataset? In that case this could be placed with the dataset. Or drop anyway and split into several tables, this would also increase comparability.
- Qian et al. (2022) two outcome measures named, three results given. No sorry, this is only poor readability of the table.
- Hosney et al. (2022) giving the averages only, also gives the feeling of an loss of information.
- Chen et al. 2022: what is RS? And what if not combined?
- Bechelli et al : what is DL (Deep Learning?) what ML (general Machine Learning Models?) this gets not clear. Either use abbreviations in the table or don't, but keeps consistent with this decisions and make sure abbr. are clear.
- avoid linebreaks within rows.
What do I learn from the table, where is the benefit? Comparability between methods gets completely lost.
So. This way, dropping up to 3 columns would perhaps allow for a restructuring of the table. Give a column for each measurement outcome result (column names vertical) (perhaps even use landscape orientation) and order by i.e. data set. This way the table would at least allow for comparison of different applications performances on the same dataset.
- I would drop figure 2 as it does not carry important information that could not be stated in 1-2 sentences.
- Line 228: it does not get clear which one are those 11 studies
Table 3: Put this table before table 1, and then drop the image type column in table 1.
Also please have a closer look at the table layout: some rows are split over pages which results in poor readability, while other entries have so much space above and beneath.
Dear reviewer, thank you for this thorough and thoughtful feedback. We have entirely revised the results section including the table formats, order, readability, and text content. We hope you find this helpful.
- Discussion again is written nicely.
However, this article could be improved so much by having a closer look. Improve the table for better comparability. I.e. Table 1: skreekala / sangers / samsudin: direct comparison is possible at least between two of them since they're close, so one could state that samsudin might be better than skreekala. BUT what is the difference between them? And is the only difference between them and sangers et al. the outcome measure used?
And maybe their work is based on the work of combalia et al. - what made the improvement in those three years between those works?
As given here, the overview is shallow.
Dear reviewer, thank you for this thorough and thoughtful feedback. We have revised our discussion section accordingly.
Reviewer 4 Report
Comments and Suggestions for Authors
Dear authors,
I evaluated the article titled “Artificial Intelligence Applied to Non-Invasive Imaging Modalities in Identification of Nonmelanoma Skin Cancer: A Systematic Review”. The goal was to “analyze the current state of the literature regarding novel artificial intelligence (AI) machine learning models utilized in non-invasive imaging for the early detection of nonmelanoma skin cancers. Furthermore, we aimed to assess their potential clinical relevance”.
The theme is current and important for the area.
INTRO
It was divided in intro and background. Therefore, I recommend to do only Introduction and adjusted the text.
M&M
- why just articles between January 2018-November 2022? Why not up to November 2023 and before 2018?
- This section is mixed with Results. Please, I suggest to split them.
- the description of this section is incomplete.
RESULTS
- For this type of study, it is lacking information in the results section.
DISCUSSION AND CONCLUSION
- both sections must be reviewed. I considered the discussion poorly performed.
Author Response
Dear authors,
I evaluated the article titled “Artificial Intelligence Applied to Non-Invasive Imaging Modalities in Identification of Nonmelanoma Skin Cancer: A Systematic Review”. The goal was to “analyze the current state of the literature regarding novel artificial intelligence (AI) machine learning models utilized in non-invasive imaging for the early detection of nonmelanoma skin cancers. Furthermore, we aimed to assess their potential clinical relevance”.
The theme is current and important for the area.
Thank you for the time taken to review our manuscript. We have reviewed your recommendations and several revisions have been made accordingly.
INTRO
It was divided in intro and background. Therefore, I recommend to do only Introduction and adjusted the text.
Dear reviewer, thank you for the time taken to provide feedback on our manuscript. We have made this change as suggested.
M&M
- why just articles between January 2018-November 2022? Why not up to November 2023 and before 2018?
- This section is mixed with Results. Please, I suggest to split them.
- the description of this section is incomplete.
Thank you for this feedback. We have adjusted the inclusion criteria to include manuscripts through December 2023. We also have ensured that the methods and results sections are clearly distinguishable.
RESULTS
- For this type of study, it is lacking information in the results section.
Upon review of reviewer suggestions, we have revised and supplemented the results section.
DISCUSSION AND CONCLUSION
- both sections must be reviewed. I considered the discussion poorly performed.
Upon review of reviewer suggestions, we have revised and supplemented the discussion and conclusion sections.
Reviewer 5 Report
Comments and Suggestions for Authors
the work is very interesting, well structured, correctly articulated, the analysis is precise, I made some observation related to the inclusion and exusion criteria of the selected papers of the review eems to me to be an excellent analysis and review of the literature
maybe I suggest to better explain the inclusion and exclusion criteria of inclusion and exclusion of papers for the review. Moreover I d like a comment to the data of graphic/diagram where is indicated that in 2023 yhere were no papers pubblicated. Why? Give us an author comment.
Author Response
the work is very interesting, well structured, correctly articulated, the analysis is precise, I made some observation related to the inclusion and exusion criteria of the selected papers of the review eems to me to be an excellent analysis and review of the literature
maybe I suggest to better explain the inclusion and exclusion criteria of inclusion and exclusion of papers for the review. Moreover I d like a comment to the data of graphic/diagram where is indicated that in 2023 yhere were no papers pubblicated. Why? Give us an author comment.
Thank you for this feedback. We have adjusted the inclusion criteria to include manuscripts through December 2023. We also have also made revisions to the methods section to improve the quality of this section.
Round 2
Reviewer 1 Report
Comments and Suggestions for Authors
The article has been revised considerably and the current version is more logical, but still with a few minor problems.
1. Theoretically, reference should not appear in Abstract.
2. The methods and conclusions mentioned in 1.1 and 1.2 lack references.
Comments on the Quality of English LanguageMinor editing of English language required
Author Response
Dear reviewer, thank you for your feedback. We originally added the reference in the abstract per another reviewer's recommendation; however, we agree with your recommendation and have since removed this reference.
We have also provided references to sections 1.1 and 1.2.
Reviewer 2 Report
Comments and Suggestions for Authors
It was difficult to read a file with 'tracking changes' on.
The paper is updated nicely and it is ok from my side.
Author Response
Dear reviewer, thank you for your feedback and support of our manuscript!
Reviewer 3 Report
Comments and Suggestions for Authors
Thank you for the changes, the paper has improved a lot. Also the table is much more helpful now. Maybe consider to mark the best results (per dataset & quality measurement value) with bold font.
Only thing that would improve the paper now imho would be to describe the key developments of the considered papers.
Author Response
Dear reviewer, thank you for your time and thoughtful feedback. We received many recommendations for the readability of the table and are pleased to hear that you find it satisfactory. In consideration of your suggestion to bold the highest values in the table, we have chosen not to in order to avoid complicating its view. Rather, we have pointed out the highest values in our texts with references that point to the respective row in our table.
Key developments of included studies can be found in our results section. Thank you for your time!
Reviewer 4 Report
Comments and Suggestions for Authors
Dear authors,
thank you for the revision sent. I considered it enough.
Congratulations!
Author Response
Dear reviewer, thank you for your time and feedback.